# AC2: An Efficient Protein Sequence Compression Tool Using Artificial Neural Networks and Cache-Hash Models

**DOI:** 10.3390/e23050530

**Published:** 2021-04-26

**Authors:** Milton Silva, Diogo Pratas, Armando J. Pinho

**Affiliations:** 1IEETA—Institute of Electronics and Informatics Engineering of Aveiro, University of Aveiro, 3810-193 Aveiro, Portugal; ap@ua.pt; 2Department of Electronics Telecommunications and Informatics, University of Aveiro, 3810-193 Aveiro, Portugal; 3Department of Virology, University of Helsinki, 00014 Helsinki, Finland

**Keywords:** lossless data compression, protein sequence compression, context mixing, neural networks, mixture of experts, coronavirus

## Abstract

Recently, the scientific community has witnessed a substantial increase in the generation of protein sequence data, triggering emergent challenges of increasing importance, namely efficient storage and improved data analysis. For both applications, data compression is a straightforward solution. However, in the literature, the number of specific protein sequence compressors is relatively low. Moreover, these specialized compressors marginally improve the compression ratio over the best general-purpose compressors. In this paper, we present AC2, a new lossless data compressor for protein (or amino acid) sequences. AC2 uses a neural network to mix experts with a stacked generalization approach and individual cache-hash memory models to the highest-context orders. Compared to the previous compressor (AC), we show gains of 2–9% and 6–7% in reference-free and reference-based modes, respectively. These gains come at the cost of three times slower computations. AC2 also improves memory usage against AC, with requirements about seven times lower, without being affected by the sequences’ input size. As an analysis application, we use AC2 to measure the similarity between each SARS-CoV-2 protein sequence with each viral protein sequence from the whole UniProt database. The results consistently show higher similarity to the pangolin coronavirus, followed by the bat and human coronaviruses, contributing with critical results to a current controversial subject. AC2 is available for free download under GPLv3 license.

## 1. Introduction

One of the most demanding challenges in data compression is related to the lossless compression of protein (or amino acid) sequences. These sequences’ origins follow the gene expression process, from DNA to RNA, to make a functional product: a protein. The first phase is transcription, where the information in every cell’s DNA, possibly non-contiguous, is converted into small, portable RNA messages. Symbolically, only the T symbol is transcripted in U from the 4-symbol DNA alphabet (A, C, G, T). The second phase is the translation, where each triplet of RNA is encoded into one of the twenty possible amino acids. Here, it is essential to remember that a different triplet can create the same amino acid and, hence, it is a lossy encoding process. Finally, a specific chain or set of chains of amino acids establishes a protein.

Although proteins have a three-dimensional (3D) structure that reshapes over time, they are usually represented in FASTA files as a (static) 1D string of characters. Therefore, structural correlations between similar parts need to be modeled to extend compression gains. These involve the beta-pleated sheets, alfa-helix, side-chain interactions, and possible combinations of multiple amino acid chains [1]. Recent developments in protein folding have shown that high structural prediction can be achieved from the DNA source [2]. However, for coding applications, time constraints and the use of side information provides additional challenges. In addition, extra challenges are presented when modeling protein sequences without information of the DNA sequence (which is the case in this article) and, hence, relying on the product of a process that may contain errors and imprecision at several phases. Therefore, data compressors need to consider protein sequences containing extra symbols to represent ambiguity, error, indecision, or incompleteness [3].

The main purpose of data compression is to reduce storage and increase transmission efficiency, specifically requiring high speed and efficient data compressors [4]. A growing application for data compression is data analysis, for example in Bioinformatics [5]. In this case, the main focus is on the maximum compression ratio, although the improvement in data compressors’ speed is valuable, particularly for large-scale data analysis [6]. As an analysis tool in the genomics and proteomics fields, data compression has been used in many applications [7], for example, to estimate the Kolmogorov complexity of sequences [8,9], classification [10,11], phylogenomic and phylogenetic analysis [12,13], information retrieval [14], variation and rearrangement detection [15,16], structural analysis [17,18], pan-genome analysis [19], metagenomics [20], detection of DNA-binding proteins [21], and domain composition studies [22,23].

In the literature, the use of specialized protein sequence compressors is low compared with that of DNA sequence compressors, mainly because specialized programs, similar to those that model the inverted repeats in DNA, are much harder to design, given the substantial higher uncertainty and lower specific sequence patterns [24]. Therefore, high ratio general-purpose data compressors are very close to the specialized category.

In the specialized category, the ProtComp [25] exploits approximate repeats and uses a hybrid method combining a substitution approach using Huffman coding and a first-order Markov model with arithmetic coding. ProtCompSecS [26], adds to ProtComp a dictionary-based method to encode the secondary information related to proteins. The algorithm presented in [27] uses the Burrows-Wheeler transform and the sorted common prefix combined with substitutions to exploit sequence long-range correlation.

In 2007, Benedetto et al. show that models that consider short and medium size correlation were more likely to achieve higher compression rates [28]. This characteristic was applied in XM through the combination of expert models with short and medium size, namely repeat and context models [29]. A fusion of dictionary and sequence alignment methods for the compression of protein databases was proposed in CaBLASTP [30]. This algorithm searches for solid sequence alignments, and when one exists, it stores an index instead of the sequence. In [22], a heuristic approach was proposed to transform a hypergraph representing the proteome into a minimum spanning hypertree. In 2017, CAD [31] was proposed, relying on an adaptive dictionary with Huffman coding.

More recently, the challenge of protein sequence compression has been revisited, namely with the proposal of AC, NAF, and CoMSA. Specifically, the AC tool [9,32] uses an ensemble of Markov models (finite context and substitution tolerant context models) with adaptive weights per model and arithmetic encoder. The NAF tool [33] uses a 4-bit encoding followed by its compression with general-purpose compressor zstd (https://github.com/facebook/zstd, accessed on 23 April 2021). The challenge of data compression in aligned data gain momentum with CoMSA [34], a compression tool using the generalization of the positional Burrows-Wheeler transform for non-binary alphabets. In the natural sequence domain, interesting approaches using prediction-based compression through the decoupling of the context model from the frequency model have been proposed [35].

In this article, we describe AC2, an evolution of the AC compressor. Contrarily to AC, AC2 uses a neural network to mix the experts and memory caches for the models with high context orders. Specifically, AC2 takes a meta-learning approach to the mixture of experts [36]. We use a neural network with input of the probabilities of each amino acid given by each Markov model. As additional inputs, we derive other features to improve the accuracy of the network. As outputs, the network uses one node per amino acid. Each output node gives the final probability for the corresponding amino acid. We use a small multilayer perceptron for the neural network, which is trained online for each new symbol in the sequence. Moreover, to reduce the model’s memory, specifically in the compression of protein sequence collections, we develop cache-hashes for the highest-order context models.

The main contributions of this paper can be summarized by the following points:An efficient and open-source protein sequence compressor (implemented in C language) for reference and reference-free modes (AC2);An enhanced neural network mixer with detailed instructions on how to integrate it into other statistical compressors;A cache-hash memory model for the deepest context orders with generic alphabets;A protein sequence compression benchmark for reference and reference-free modes;Similarity analysis for each reference SARS-CoV-2 protein sequence (host: *H. sapiens*) according to all the existing UniProt viral protein sequences.

The next sections of this article present the implementation details of AC2, the comprehensive benchmark results, including several protein sequences with different characteristics, and state-of-the-art compressors. Additionally, we provide compression-based analysis examples employing the AC2 compressor.

## 2. Methods

This section presents the details of the AC2 compressor methodology. AC2 uses identical models as AC, namely a combination of context models and substitution tolerant context models of several order depths. The usage of substitution tolerant models in biological sequences is crucial because they provide a solid improvement factor over high-ratio general-purpose data compressors and, hence, are models that can be considered of specific biological nature [37,38].

The most significant developments of AC2 are a neural network to augment the expert mixing and an individual memory cache-hash for the models with the highest context orders. These implementations allow AC2 to improve the compression ratio while reducing the memory usage substantially. In the following subsections, we provide details on the neural network, cache-hash and counter precision, and parameters used.

### 2.1. Selecting a Neural Network Type

In this subsection, we review the literature related to selecting suitable artificial neural networks for sequence prediction, namely, to incorporate an appropriate network into the proposed data compressor. The computational resources required and the feasibility of the network integration in the data compressor are essential because we are concerned with the network accuracy and efficiency.

Although we searched explicitly for literature that compared different architecture networks in protein sequence compression, we were unable to find a single work. Nevertheless, we found an analogous work, specifically, DNA sequence compression [39]. This analogous work describes a compressor based on neural networks and compares two types of recurrent neural networks (RNNs) and a multilayer perceptron (MLP). Overall, the RNNs provide the best compression rate, although there is a dependency on the dataset. Moreover, this method uses a top-of-the-line GPU, consuming several hours to compress sequences with 10MB length. Because performing a compression-based analysis in an extensive protein sequence dataset (with gigabyte length) is one of the applications of our work, speed is critical; therefore, this limits the usage of this methodology. We also found benchmarks of neural networks to time series prediction problems; these seem to fit with the stochastic nature of the issue we are analyzing [40].

In terms of computational resources, we intuitively deduce that the multilayer perceptron (MLP) would have the best performance because of its elementary nature; this is supported by [41], where we can notice that even with large networks, the MLP is the fastest network by approximately 50%, and the convolutional is the second fastest, followed by the recurrent networks.

In terms of accuracy, the performance appears to be very dependent on the dataset. In [42], a multilayer perceptron (MLP), a convolutional neural network (CNN), a recurrent neural network (RNN), and a long short term memory (LSTM) are used to predict the values of the stock market. The results favor the CNN followed by the MLP, with the LSTM and the RNN trailing behind. In diverse datasets, the MLP is superior to the CNN. In [41], an MLP, a CNN, and an LSTM are compared using several datasets. Overall, the CNN performs best, followed by the LSTM and then the MLP. As in the previous comparison, there is no single network that achieves the best result in all datasets. In [43], the authors compare an LSTM to an MLP and conclude that the MLP has performance equal to or better than that of the LSTM. In [44], several neural networks and datasets are compared; among these are various types of CNNs and an MLP. Two types of CNNs, the residual neural network (ResNet) [45] and the fully convolutional neural network (FCN) [46], provide the best overall accuracy, with the MLP placing fourth out of the nine evaluated networks. In [47], a hybrid network combining a CNN and an LSTM is used to predict power consumption, stock values, and gas concentrations. In some datasets, the CNN has better predictions than the LSTM. The proposed hybrid approach always presents better predictions in the three datasets. In [48], two neuro-fuzzy networks are compared with an MLP to model the reference evapotranspiration. The neuro-fuzzy networks displayed higher accuracy than the MLP. Finally, in [49], we see a comparison of a neuro-like structure with a sequential geometric transformations model (SGTM) [50] and an MLP. The SGTM has better accuracy and spends less time during the training phase.

No network appears to be the best in all datasets, as anticipated by the results of the *no free lunch* theorem [51].

### 2.2. Neural Network Architecture

Based on the above literature review, we elected to use the MLP, which confers accurate and efficient predictions [42]. Specifically, this network is one of the most resource-efficient neural networks in execution time and memory usage. Furthermore, it has demonstrated high performance in other tasks also using biological sequences [52] and is straightforward to implement and validate.

Specifically, the network has a single hidden layer, where all nodes have the sigmoid activation function. The output layer uses the softmax activation, ensuring that all output nodes sum to one, and the loss function is the cross-entropy. The network has two bias nodes, one in the input layer and one in the hidden layer. The weights are set according to the Xavier initialization [53].

### 2.3. Neural Network Inputs

Figure 1 depicts a high-level overview of the mixer architecture, including its inputs.

The inputs to the network consist of the outputs of the context models and substitution tolerant context models and the output of the mixing done in the AC compressor. Therefore, we do not substitute the mixing done in AC, but instead, we are augmenting it. In other words, the mixing performed in AC is considered as another model. We transform these probabilities by subtracting $\frac{1}{n}$, where *n* is the number of unique symbols in the sequence. After the subtraction, we multiply the result by five in the case of a model and ten in the AC mixer output case. These types of transformations and their motivation are explained in [54].

Moreover, we use derived features with the mean symbol frequencies for the last 8, 16, and 64 symbols; these are also multiplied by five. Finally, we use an exponential moving average for all symbols, such that when a symbol occurs, the average for symbol *i* is updated according to
(1)$$avg_{i}0.8+0.2*avg_{i}.$$

If symbol *i* does not occur, then the update rule is
(2)$$avg_{i}0.2*avg_{i}.$$

### 2.4. Neural Network Outputs and Training

For the outputs, we use one node per amino acid; each one has the probability for that symbol. These are the values that are passed to the arithmetic encoder. The network is trained online for each new symbol. The training vector is filled with zeroes except for the position corresponding to the symbol that occurred. This position has a value of one. The training algorithm is the stochastic gradient descent without momentum [55].

### 2.5. Neural Network Pre-Training and Selection Heuristic

We noticed that the neural network initialized with the random values had less accurate predictions at the beginning of the sequence. We improved the network, pre-training it by activating the same symbol in all models with a value of one. Moreover, the same symbol is used for training. Essentially, this forces the bias that if all models agree on the same symbol with absolute certainty, then the output is forced to be that symbol.

Additionally, we added a heuristic that selects between the AC mixer and the neural network output. The mixer used is the best performing one. This choice is determined by an exponential moving average of the number of estimated bits produced by the two mixers.

### 2.6. Cache-Hash and Counter Precision

One of the significant limitations of AC is the substantial increase of RAM provided with the combination of high-context orders and large sequence sizes. For example, when the sequences to compress are larger than, say, 200 MB and AC use context orders higher than 7, the RAM increases to values that regular laptops can not support. To resolve this issue, AC2 uses cache-hash memory models.

A cache-hash [56] enables storing in memory only the latest entries up to a certain number of hash collisions. This model enables the use of deep context orders with very sparse representations. If AC2 stored its entries in a table, this would require ${\left|A\right|}^{k+1}$ entries, where |A| is the size of the protein sequence alphabet and *k* the context order of the model; this means that assuming counters of 8 bits for a *k* = 10 and an |A| of 20 would require 186 TB of RAM. For small sequences, a linear hash would be feasible, depending on the available RAM. However, for large sequences, this becomes unfeasible.

Therefore, AC2 uses a cache-hash for each high context order model to remove space constraints. AC2 uses a parameter that represents the maximum number of collisions, enabling a constant maximum peak of RAM. Moreover, we reduced the size of the counters for the models. AC2 now uses two bits per symbol, unlike eight bits in AC. The practical outcomes are speed and RAM improvements, enabling the compression of extensive sequences, specifically large collections of protein sequences. The disadvantage is the slightly higher code complexity of AC2.

### 2.7. Parameters and Optimization

In addition to the AC parameters, AC2 includes parameters to control the learning rate, the number of nodes in the hidden layer, and the cache’s size per model. AC2 also adds a more powerful compression level.

All the internal and external parameters were determined empirically. These include the coefficients for the exponential moving averages, the window size for the moving averages, the input transformations, the learning rates, the hidden layer, and the number of models and their parameters. The internal parameters are fixed for all experiments, while the external parameters were adjusted for each sequence; the parameters used are available in the same repository as the source code.

## 3. Results

In this section, we evaluate the performance and accuracy of AC2 as a protein sequence compressor in two benchmarks, namely in reference-free and reference-based compression. AC2 is available for free download (GPv3 license) at https://github.com/cobilab/ac2, accessed on accessed on 23 April 2021.

### 3.1. Datasets and Materials

For benchmarking AC2 as a reference-free compressor, we used two datasets, namely

**DS1**: three protein databases used in [57]:-UniProt: the UniProt collection of sequences [58];-PDBaa: the Protein Data Bank [59];-GRCh38: the human reference genome [60].**DS2**: a comprehensive dataset (proposed in [9]), containing the following sequences:-BT: Bos taurus;-HS: Homo sapiens;-SC: Saccharomyces cerevisiae;-HT: Haloterrigena turkmenica;-EC: Escherichia coli;-LC: Lactobacillus casei;-SA: Staphylococcus aureus;-HI: Haemophilus influenzae;-MJ: Methanococcus jannaschii;-DA: Desulfurococcus amylolyticus;-AP: Acanthamoeba polyphaga;-HA: Hadesarchaea archaeon;-FM: Fomitiporia mediterranea;-FV: Fowlpox virus;-XV: Xanthomonas virus Xp10;-EP: Enterococcus phage.

For benchmarking AC2 as a reference-based compressor, we used four complete proteomes of four primates (human, gorilla, chimpanzee, and orangutan) with a pairwise chromosomal compression. For each chromosomal pair, the following compression was performed:Chimpanzee (PT) using human (HS) as a reference;Gorilla (GG) using human (HS) as a reference;Orangutan (PA) using human (HS) as a reference.

Unless otherwise stated, the benchmarks were performed on an Intel(R) Core(TM) i7-6700 CPU @ 3.40GHz running Linux 5.4.0 with the scaling governor set to performance and 32GB of RAM.

### 3.2. Reference-Free Compression Benchmark

For the comparison with AC2, we selected two specialized protein compressors: AC and NAF. These are the only available and working compressors we could find; therefore, we added general-purpose compressors to make a comprehensive comparison. The general-purpose compressors added are the Big Block BWT (Burrows-Wheeler transform) [61], the LZMA [62], and the CMIX [63].

The results from Table 1 show that AC2 achieves the best compression ratio for all sequences, except for the smallest sequence (EP). For DS1 (collections of protein sequences), AC2 achieves gains of 2%, 4%, 7%, 9%, and 35%, compared to CMIX, LZMA, NAF, AC, and BBB, respectively. For DS2 (individual proteome sequence compression), AC2 achieves gains of 2%, 3%, 4%, 6%, and 13%, compared to AC, CMIX, NAF, LZMA, and BBB, respectively.

It should be noted that the bits per symbol (bps) are for the resulting archive file, which in the case of AC2 includes a header with parameters to describe the models and the network parameters. For small sequences, this is a significant portion of the final size. If we ignore the header size, which makes sense when an analysis is a primary goal, AC2 achieves even more significant gains than AC. Even with this disadvantage, AC2 shows a better compression ratio than AC for all sequences tested.

In terms of memory usage, AC2 uses substantially less RAM than AC, and the cache size parameters limit the increase in RAM. In challenging cases, such as the UniProt sequence’s compression, the memory usage of AC2 is approximately 86% less than that of AC (111 GB to 16 GB). With the present configuration, the AC2 uses less memory than the two closest higher compression ratio compressors, CMIX and AC. For the UniProt sequence, the RAM usage is 660 MB, 740 MB, 3 GB, 16 GB, 19 GB, and 111 GB for LZMA, BBB, NAF, AC2, CMIX, and AC, respectively. Parameters control the memory requirements of AC2; thus, the needed memory can be decreased with a penalty of a small precision payoff, providing utility to computers with lower computational resources.

The computational time of AC2 is ≈3× slower than of AC, but it is ≈7–19× faster than CMIX. Compared to the other compressors (BBB, LZMA, and NAF), AC2 is ≈15–49× slower. Compressing the UniProt sequence with AC had to be done in a different machine with more RAM but a slower processor; this is why the execution was slower. While AC2 is slower than AC, this gap should decrease soon due to the inclusion of specialized instructions and data types in general-purpose consumer CPUs [64,65]. The parameters used for this benchmark focus on maximum compression ratio, but we can decrease the execution time while maintaining the best ratios. For example, in the PDBaa sequence, we can reduce the number of hidden nodes from 128 to 40, which gives us 1.725 bps at half the execution time (10 min).

Figure 2 depicts the difference between the AC and the AC2 compression performance, described as gain complexity profiles. Gain complexity profiles are numerical representations of the gain in terms of bits per symbol for each sequence element. Using the GTO toolkit [66], we applied a low-pass filter to the gain complexity profiles to smooth the peaks and valleys and better perceive the trends.

We can see the heuristic effect of switching between the new and the old mixer for small sequences, such as the XV and FV. Flat regions, with the number of bits per symbol equal to zero, corresponds to regions where the old mixer is used. It is also visible for smaller sequences that even with the heuristic and the pre-training, the AC2 mixer sporadically produces lower results than the AC mixer. These are shown in the plot when the graph has negative values. This situation is due to the lag associated with the exponential moving average that controls the mixer switching. The lag can be reduced, and for small sequences, the compression does benefit; the reverse is true for large sequences. For extensive sequences (HS), we can see that the plot is always positive with this smoothness level. In this case, AC2 appears to compress consistently more.

Finally, all plots show small peaks of at most 0.4 Bps. On the one hand, this is due to the smoothness function applied. On the other hand, it shows no large regions (as a percentage of the total sequence) where AC2 is vastly superior to AC. Even so, there appear to be new regions of interest that could provide new insights into the sequences’ nature.

### 3.3. Reference-Based Compression Benchmark

In this section, we compare AC with AC2 for the compression of proteins using a reference because, as far as we know, AC it is the only data compression tool (currently working) for reference-based protein sequence compression. AC has some reference-based compression errors related to the incapability to deal with a different alphabet between reference and target sequences. Therefore, we improved AC2 to output the AC compression estimates, using the AC mixer’s probabilities to calculate the expected number of bits with $-\log_{2}\left(p_{sym}\right)$. The results in Table 2 show that AC2 improves the compression ratio by 6–7% compared to AC. Chromosomes 5 and 17 of the gorilla show the least improvement and also the worst bits per symbol. This performance is due to a hypothetical rearrangement in the gorilla that diverges from the other three primates [67]. In practice, a similar part of one of the chromosomes is present in the other, decreasing the capability of using the reference sequences as an auxiliary input [68]. A way to improve these particular sequences’ compression would be to use the references of both proteomic sequences from chromosomes 5 and 17 of the human.

The plots in Figure 3 show a similar trend to the plots from the reference-free compression. For the mitochondrion sequence, we can see the effects of the heuristic switching between the AC and neural network mixer. For the larger sequences, we can see more consistently positive values.

## 4. Application: SARS-CoV-2 Protein Sequence Similarity to Other Viral Proteins

As an example of identifying similar protein sequences in terms of quantity of information, we studied the most similar protein sequences, in the whole UniProt database, to the proteins of the human *Severe acute respiratory syndrome coronavirus 2* (SARS-CoV-2) [69], respecting important Bioinformatics guidelines [70].

SARS-CoV-2 is a positive-sense single-strand RNA virus with an origin traced to a food market in Wuhan, China, in December 2019 that can cause COVID-19 disease [71]. SARS-CoV-2 is transmitted by inhalation or contact with infected droplets with an incubation period from 2 to 14 days [72]. According to the World Health Organization (WHO), SARS-CoV-2 has already caused more than 146 million infections and 3.1 million deaths, where the variation of the latter seems to be related to seasonality [73]. New developments of fast diagnostic methods are emerging, for example, a 10-min antibody assay [74], enabling reacting and predicting infection and vaccine responses much faster.

Although several therapeutics have already been proposed [75,76], the emergence of multiple variants brings additional complexity to the challenge both in diagnostics and therapeutics [76,77]. Much has been learned with the current pandemic; however, much progress is still required. One of the inconclusive themes is related to SARS-CoV-2 protein sequence similarity to other viral protein sequences. Despite several studies addressing this topic both at genomic and proteomic level [78,79], different interpretations have been provided at the animal host origins [80,81,82]. Perhaps the reason is related to the characteristics of the measures used, namely the use of normalized scores applying alignments that do not rigorously consider quantities of information, overestimation issues, and the respect of distance properties, such as symmetry and idempotency [83]. Without respecting the theoretical foundations that characterize an information distance (or distance of dissimilarity), a problematic question arises: where should one draw the threshold or the splitting line? Therefore, defining thresholds in these cases can substantially contradict conclusions, not because one of the measures is incorrect but because the starting point or assumptions are fundamentally different.

Accordingly, in this paper, we present the SARS-CoV-2 protein sequence similarity to other viral protein sequences relying on AC2, the data compressor benchmarked with the best-known compression ratios (shown in this paper above) and through the computation of the Normalized Compression Distance (NCD) [84].

For this purpose, we separated all protein sequences in the UniProt database and all proteins of the SARS-CoV-2 into different classes. Then, we measured the (dis)similarity across each pair of elements of the classes using the NCD through the approximation of the conditional complexity [85] as
(3)$$\mathrm{NCD}\left(x,y\right)=\frac{\max\{C(x|y),C(y|x)\}}{\max\{C(x),C(y)\}}.$$

For approximating the complexity (C(x) and C(y)) and conditional complexity (C(x|y) and C(y|x)), we used AC2 with optimized parameters for each type of compression as follows:C(x), C(y): -tm 1:1:1:0.9/1:1:0.9 -tm 3:10:1:0.95/2:10:0.96 -tm 5:200:5:0.98/4:50:0.95;C(x|y), C(y|x): all the models used in C(x) and C(y) and -rm 1:1:1:0.9/1:1:0.9 -rm 3:10:1:0.95/2:10:0.96 -rm 5:200:5:0.98/4:50:0.95 -rm 7:1500:5:0.955/6:50:0.945.

Figure 4 depicts the results with the lowest NCD and, hence, the most similar sequences according to a reference SARS-CoV-2. As presented in the architecture of Figure 4d,g, several protein sequences can be localized, namely the (Open Reading Frame) ORF1ab, spike (S), envelope (E), membrane (M), and nucleocapsid (N). The ORF1ab includes ORF1a and ORF1b, which characterize a non-structural polyprotein involved in the transcription and replication of viral RNAs, containing the proteinases responsible for the protein’s cleavages. ORF3 is an accessory protein specialized for environment change inside the infected cell, through the membrane’s rupture, increasing the virus replication. The membrane (M) is a structural protein that forms part of the virus’s outer coat, playing a crucial role in virus morphogenesis and assembly via its interactions with other viral proteins. The nucleocapsid protein (N) is a structural protein that packages the RNA into a helical ribonucleocapsid (RNP) and is essential during assembly through its interplays with the viral genome and membrane protein (M). It also magnifies the efficiency of subgenomic viral RNA transcription and replication [86].

According to the remain of Figure 4, the NCD for the flagged proteins is consistently lower for pangolin coronavirus, followed by the ranked alternation between multiple bats and human coronavirus. In humans, the highest similarities stand for MERS [87] and SARS [88] coronaviruses, naturally showing the evolution under the host. The pangolin and bat coronaviruses with higher similarities to the SARS-CoV-2 are in accordance with some studies of both origin species of SARS-CoV-2 [89], measured at the genomic [78] and proteomic level [79]. Moreover, the results show that the pangolin coronaviruses are the most similar in terms of information (Kolmogorov complexity [90]). Furthermore, the last protein sequence (marked with an X in Figure 4), also known as ORF10, shows only relevant similarity according to the pangolin coronavirus. Despite the consistency of the results provided at the proteomic level, the discovery of new proteomes with higher similarity (or lower NCD) to the SARS-CoV-2 may change the conclusions.

## 5. Conclusions

This paper describes AC2, a new protein sequence compressor that uses a neural network to mix experts with a stacked generalization approach and individual cache-hash memory models to the highest context orders. We show gains over the previous compressor (AC) between 2 and 9%, depending on the dataset characteristics. These gains come at the cost of slower execution times ≈3×. AC2 substantially improves the memory usage compared to AC, with memory usages about 7× lower. Compared to the previous best available state-of-the-art compressors, AC2 achieves an overall compression ratio improvement of approximately 2% and 6% in reference-free and reference-based modes, respectively. Nevertheless, we think that AC2 can still be improved in single-organism proteome compression. For example, to address this challenge, we can derive other experts that model the secondary information of the proteins, similar to the algorithm in [26]. Another crucial area of improvement has to do with the computational resources, as these may limit the efficiency of analysis. To improve the execution speed, we can drive computations to a GPU, with the neural network as the most likely candidate to benefit. Furthermore, different caching strategies directly applied to the models may reduce the memory requirements while bringing some improvements.

Additionally, we provided an application of the usage of AC2 for comparative proteomic analysis, namely measuring the similarity between each SARS-CoV-2 viral protein sequence with each viral protein sequence from the whole UniProt database. This straightforward alignment-free solution infers the most similar proteomic sequence using very flexible, balanced, and consistent measures. According to the eventual redundancy in the sequences, alignment-based measures may provide overestimated results, given its small size and ambiguous choices. On the other hand, our approach quantifies the similarity using information without overestimation (a property of using data compression through the NCD). Moreover, it uses multiple experts of different nature in an unsupervised learning approach. This characteristic means that the data compressor can use models of another nature, for example, energy, structural, or algorithmic models [91], to combine the predictions besides simple vertical amino acid comparison. In this paper, the results consistently show higher similarity to the pangolin coronavirus in the provided application, followed by the bat and other human coronaviruses. However, as with any other known comparative methods, this approach has a drawback: discovering new proteomes with higher similarity to the SARS-CoV-2 may change the conclusions.

## Figures and Tables

**Figure 1 entropy-23-00530-f001:**
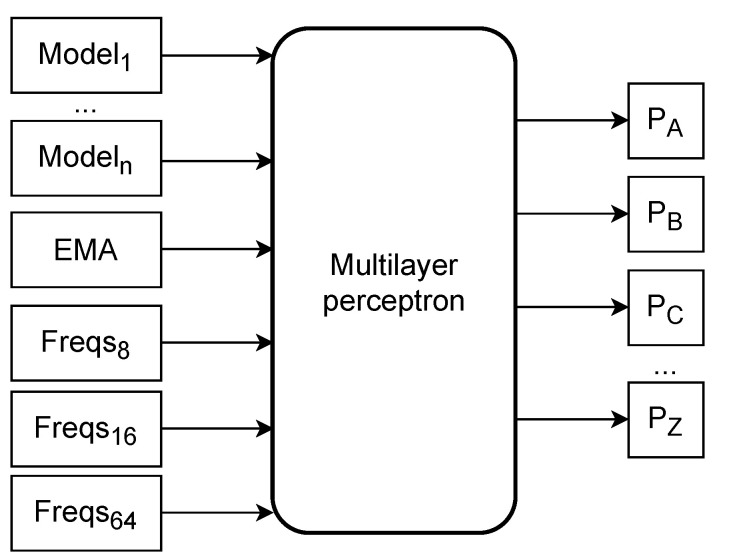
Mixer architecture: high level overview of inputs to the neural network (mixer) used in AC2. *Model*_1_ through *Model_n_* represent the AC model outputs (probabilities for all the amino acids). *EMA* represents the exponential moving average for each symbol. *Freqs* are the frequencies for the last 8, 16, and 64 symbols. The network outputs represent the probabilities for each amino acid symbol.

**Figure 2 entropy-23-00530-f002:**
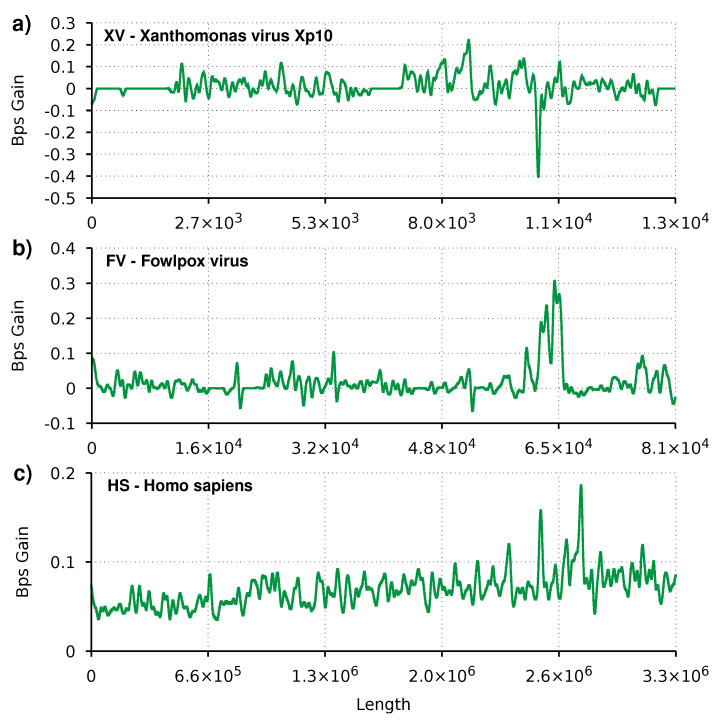
Smoothed gain of AC2 relatively to AC, in bits per symbol (bps). Regions with the line above zero indicate that AC2 has better compression than AC. Three profiles are depicted for three species, namely (**a**) XV: *Xanthomonas* virus Xp10, (**b**) FV: *Fowlpox* virus, (**c**) HS: *Homo sapiens*.

**Figure 3 entropy-23-00530-f003:**
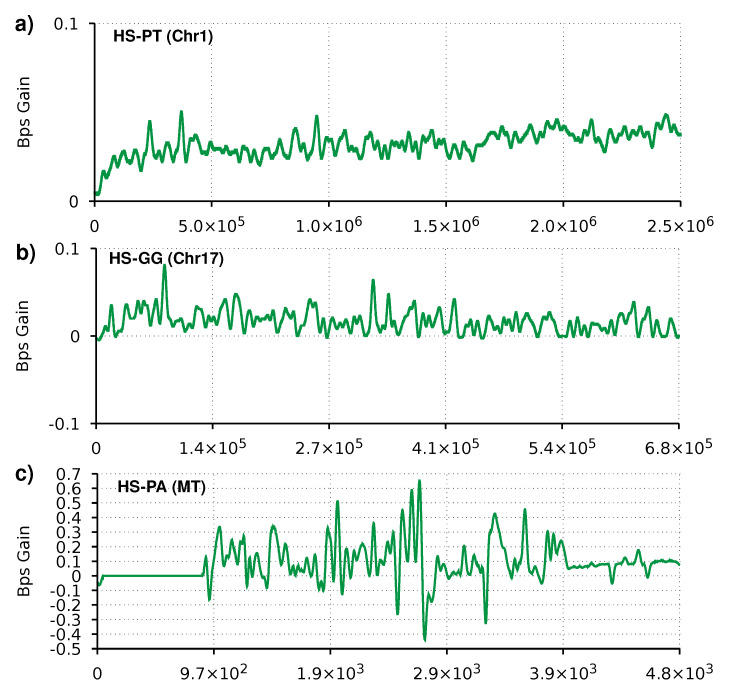
Smoothed gain of AC2 relatively to AC in bits per symbol (Bps). Regions with the line above zero indicate that AC2 has better compression than AC. Three profiles are depicted for referential compression of three sequence pairs, namely (**a**) Chromosome 1 of chimpanzee, (**b**) Chromosome 17 of gorilla, and (**c**) Mitochondrion of orangutan. All target sequences use the corresponding human sequence as reference. The compression parameters are the same as in Table 2.

**Figure 4 entropy-23-00530-f004:**
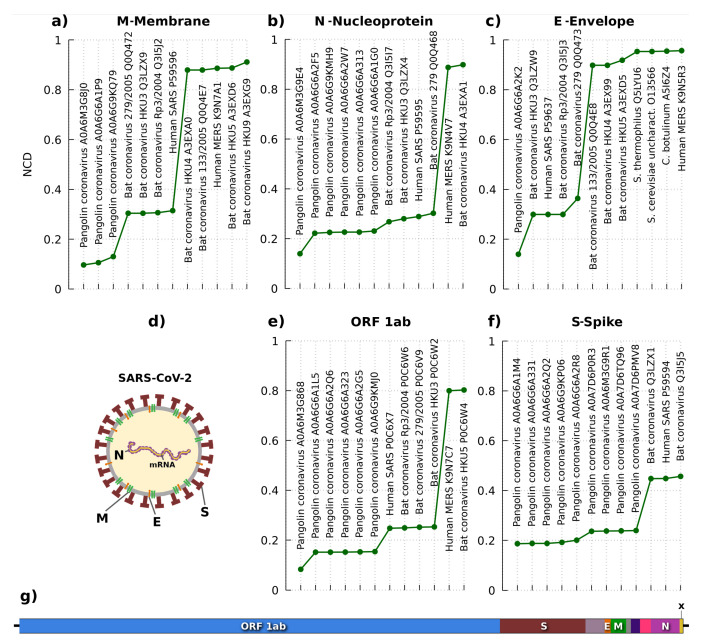
Analysis of the most similar protein sequences from the NCBI database according to multiple protein sequences of the SARS-CoV-2. The similarity metric used is the Normalized Compression Distance (NCD). The lower the NCD, higher the similarity. Five protein sequences are used for comparison: (**a**) membrane, (**b**) nucleoprotein, (**c**) envelope, (**e**) Replicase polyprotein (ORF 1ab), and (**f**) spike. The (**d**) panel depicts an illustration of the two-dimensional localization of the proteins in SARS-CoV-2, while (**g**) shows localization in one-dimension of the sequences that correspond to the proteins.

**Table 1 entropy-23-00530-t001:** The bits per symbol (bps) and time needed to represent an amino acid sequence for BBB, LZMA, CMIX, NAF, AC and AC2. NAF uses the highest compression level (22) for all sequences. BBB uses the parameters ‘cfm100q’ for all sequences. LZMA uses the highest level (-9 -e) for all sequences. For DS2, AC and AC2 use the same levels as in [9]. For DS1, the models used by AC are ‘-tm 1:1:0.76/1:1:0.88 -tm 2:10:0.83/1:1:0.86 -tm 3:20:0.83/2:1:0.87 -tm 4:50:0.88/2:10:0.89 -tm 5:200:0.94/3:20:0.89 -tm 6:300:0.91/5:20:0.88 -tm 7:500:0.91/6:60:0.87 -tm 8:500:0.92/7:15:0.89 -tm 9:1000:0.92/8:15:0.9 -tm 10:1500:0.92/9:80:0.9 -tm 11:1500:0.93/10:200:0.92 -tm 12:1500:0.94/11:200:0.93 -tm 13:1500:0.96/12:30:0.92 -tm 14:1750:0.95/13:150:0.93 -tm 15:2000:0.94/14:250:0.92 -tm 17:2200:0.95/16:350:0.93 -tm 20:2500:0.96/19:500:0.95’, which are equivalent to level 8 for AC2. The asterisk (*) next to the time means that the compression was run on a different machine, with more RAM but slower CPUs. The underlined values represent the fastest computations, and the bold stand for the best compression rates.

		BBB		LZMA		CMIX		NAF		AC		AC2	
DS	ID	bps	Time	bps	Time	bps	Time	bps	Time	bps	Time	bps	Time
1	UniProt	2.872	2 m 34 s	1.939	3 m 20 s	1.887	47 h 08 m 07 s	2.013	3 m 26 s	2.071	2 h 03 m 37 s *	**1.857**	2 h 24 m 51 s
	PDBaa	2.541	25 s	1.851	28 s	1.847	7 h 06 m 24 s	1.824	34 s	1.790	6 m 55 s	**1.718**	20 m 09 s
	GRCh38	1.906	29 s	1.196	22 s	1.168	9 h 24 m 44 s	1.203	35 s	1.216	6 m 38 s	**1.154**	26 m 22 s
	Total	2.689	3 m 28 s	1.817	4 m 10 s	1.774	63 h 39 m 16 s	1.869	4 m 36 s	1.910	2 h 17 m 12 s *	**1.743**	3 h 13 m 00 s
	BT	3.711	9 s	3.208	10 s	3.081	2 h 54 m 59 s	3.114	12 s	3.049	1 m 35 s	**2.961**	4 m 21 s
2	HS	4.076	2 s	4.022	2 s	3.859	44 m 36 s	3.905	2 s	3.786	21 s	**3.717**	49 s
	SC	4.093	2 s	4.030	1 s	3.914	38 m 59 s	3.956	2 s	3.876	16 s	**3.835**	47 s
	HT	4.006	2 s	3.971	1 s	3.867	19 m 42 s	3.929	1 s	3.825	10 s	**3.764**	26 s
	EC	4.150	1 s	4.209	1 s	4.051	17 m 51 s	4.108	1 s	4.038	6 s	**4.000**	18 s
	LC	4.129	1 s	4.188	0 s	4.051	10 m 59 s	4.119	1 s	4.055	4 s	**4.019**	9 s
	SA	4.142	1 s	4.213	0 s	4.036	10 m 52 s	4.109	1 s	4.056	4 s	**4.008**	9 s
	HI	4.155	1 s	4.239	0 s	4.087	6 m 59 s	4.155	1 s	4.102	1 s	**4.082**	4 s
	MJ	4.059	0 s	4.141	0 s	3.974	6 m 15 s	4.069	1 s	3.997	1 s	**3.962**	3 s
	DA	4.083	0 s	4.182	0 s	4.014	5 m 31 s	4.101	1 s	4.028	1 s	**4.008**	4 s
	AP	4.084	0 s	4.106	0 s	3.951	4 m 46 s	4.073	1 s	3.985	1 s	**3.936**	5 s
	HA	4.122	0 s	4.214	0 s	4.081	3 m 03 s	4.145	0 s	4.082	0 s	**4.071**	2 s
	FM	3.968	0 s	3.508	0 s	3.538	2 m 19 s	3.537	1 s	3.426	1 s	**3.372**	3 s
	FV	4.130	0 s	4.176	0 s	4.063	1 m 12 s	4.118	1 s	4.063	0 s	**4.049**	1 s
	XV	4.188	0 s	4.258	0 s	4.140	14 s	4.176	0 s	4.137	0 s	**4.134**	0 s
	EP	4.262	0 s	4.434	0 s	**4.228**	7 s	4.348	1 s	4.323	0 s	4.314	0 s
	Total	3.900	19 s	3.642	15 s	3.507	48 m 30 s	3.553	29 s	3.476	2 m 48 s	**3.408**	7 m 27 s

**Table 2 entropy-23-00530-t002:** The bits per symbol (b) needed to represent a protein sequence of a chromosome (Chr: Chromosome, MT: Mitochondria) using as a reference the corresponding Human’s. AC and AC2 use the following models ‘-rm 1:1:1:0.9/1:1:0.9 -rm 3:10:1:0.95/2:10:0.96 -rm 5:200:5:0.98/4:50:0.95 -rm 7:1500:5:0.955/6:50:0.945 -rm 15:2000:15:0.94/14:250:0.92 -tm 1:1:1:0.9/1:1:0.9 -tm 3:10:1:0.95/2:10:0.96 -tm 5:200:5:0.98/4:50:0.95’. Additionaly, AC2 uses the following parameters ‘-lr 0.03 -hs 32’.

	Chimpanzee (PT)			Gorilla (GG)			Orangutan (PA)		
ID	AC (b)	AC2 (b)	Gain (%)	AC (b)	AC2 (b)	Gain (%)	AC (b)	AC2 (b)	Gain (%)
Chr1	0.385	**0.353**	8.3	0.420	**0.387**	7.7	0.532	**0.493**	7.3
Chr2	0.359	**0.329**	8.3	0.355	**0.326**	8.1	0.483	**0.447**	7.5
Chr3	0.377	**0.347**	7.9	0.404	**0.375**	7.1	0.474	**0.440**	7.2
Chr4	0.384	**0.359**	6.4	0.410	**0.387**	5.5	0.487	**0.458**	6.0
Chr5	0.384	**0.355**	7.6	1.598	**1.581**	1.1	0.505	**0.471**	6.7
Chr6	0.398	**0.369**	7.2	0.423	**0.394**	6.7	0.527	**0.492**	6.7
Chr7	0.433	**0.409**	5.5	0.450	**0.426**	5.4	0.586	**0.556**	5.3
Chr8	0.405	**0.378**	6.7	0.423	**0.399**	5.7	0.549	**0.519**	5.4
Chr9	0.382	**0.355**	7.0	0.411	**0.386**	6.1	0.566	**0.537**	5.2
Chr10	0.391	**0.368**	6.0	0.417	**0.394**	5.5	0.580	**0.549**	5.4
Chr11	0.399	**0.365**	8.3	0.412	**0.379**	8.1	0.573	**0.532**	7.1
Chr12	0.354	**0.324**	8.5	0.381	**0.351**	7.8	0.457	**0.422**	7.5
Chr13	0.410	**0.389**	5.0	0.438	**0.419**	4.4	0.511	**0.488**	4.5
Chr14	0.379	**0.350**	7.5	0.396	**0.369**	6.8	0.528	**0.494**	6.4
Chr15	0.362	**0.333**	7.9	0.388	**0.360**	7.1	0.485	**0.449**	7.4
Chr16	0.409	**0.375**	8.3	0.428	**0.394**	7.9	0.569	**0.528**	7.2
Chr17	0.358	**0.330**	7.9	1.476	**1.460**	1.1	0.573	**0.538**	6.0
Chr18	0.368	**0.346**	5.8	0.401	**0.378**	5.7	0.553	**0.527**	4.7
Chr19	0.446	**0.411**	7.9	0.468	**0.433**	7.5	0.667	**0.618**	7.3
Chr20	0.415	**0.390**	6.1	0.436	**0.411**	5.8	0.614	**0.586**	4.5
Chr21	0.449	**0.430**	4.3	0.476	**0.456**	4.2	0.681	**0.658**	3.4
Chr22	0.419	**0.391**	6.8	0.424	**0.397**	6.4	0.612	**0.581**	5.0
ChrX	0.486	**0.463**	4.6	0.505	**0.485**	3.9	0.580	**0.555**	4.2
MT	0.662	**0.631**	4.7	0.574	**0.525**	8.6	0.998	**0.920**	7.8
Mean	0.409	**0.381**	6.8	0.521	**0.495**	6.0	0.570	**0.536**	6.1

## Data Availability

The data and scripts used are available at https://github.com/cobilab/ac2/tree/ref-comp/benchmark-ref, accessed on 23 April 2021 and https://github.com/cobilab/ac2/tree/master/benchmark, accessed on 23 April 2021.

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
