# Peer review of "AC2: An Efficient Protein Sequence Compression Tool Using Artificial Neural Networks and Cache-Hash Models"

_entropy, 2021, doi:10.3390/e23050530_

Round 1
Reviewer 1 Report
Paper deals with compression task. Authors proposed to use ANN for this aim and show the efficiency of such an approach. Application problem is protein sequence data compression.
Paper has scientific novelty and great practical value.
It has a logical structure. Paper is technically sound.
The proposed approaches are logical, results are clear.
Abstract section is very good!
Suggestions:
- It would be good to add point-by-point the main contributions in the end of the Introduction section
- It would be good to add the reminder of this paper
- Authors should add the Related works section. They should analyse existing ANN-based approaches for solving the stated task. It would be good to take into account non iterative SGTM neural-like structure and its modifications that are more precisely and faster than MLP (DOI: 10.1007/978-3-319-91008-6_58) among others
- Conclusion section should be extended using: 1) limitations of the proposed approach; 2) prospects for the future research.
- A lot of references are outdated. Please fix it using 3-5 years old papers in high-impact journals.
Reviewer 2 Report
1) Why did the authors exploit sigmoid function in the neural network?
2) What about the cross-entropy as loss function? What was the idea that help the authors to choose this function?
3) When the authors describe the neural networks used for the intended objectives, it would be better to also mention neuro-fuzzy networks. This would allow us to open wider scenarios as these networks are particularly useful in cases where the available data are affected by uncertainty and / or imprecision. In this regard, it is advisable to include the following scientific papers in the bibliography:
- 10.1007/s11600-020-00446-9
- M. Cacciola, D. Pellicanò, G. Megali, A. Lay-Ekuakille, M. Versaci, F.C. Morabito, "Aspects about air pollution prediction on urban environmnent", 4th IMEKO TC19 Symposium on Environmental Instrumentation and Measurements 2013: Protection Environment, Climate Changes and Pollution Control; Lecce; Italy; 3 June 2013 through 4 June 2013; Code 102275
- 10.1016/j.compag.2020.105358
4) Are (1) and (2) original formulas? If not, the source should be cited.
5) How was (3) obtained?
6) The conclusions appear poor in content. Perhaps some additional comments could enrich the content.
Round 2
Reviewer 1 Report
Dear Authors,
thank you for the improvement of your paper. However, you did not respond to my suggestions.
First of all, there is no any time constraints and you must justify your choice of neural network type by at least reviewing and analyzing the different types of neural networks in the Related works section including SGTM neurak-like structure.
The scientific novelty of the paper must be clear and understandable. Each item should contain: what you did, based on what and what it resulted in. You only have the first item.
Related works section should not be half a page. It should be much larger. Please correct this shortcoming. Maybe then the problem with outdated quotes will disappear
Regarding outdated literature. I do not understand how you want to confirm the relevance of the research by quoting works from 1951, 1965, 2000, 2002 and so on. Unfortunately, I don't have to look for literature, you do. If you do not fix this problem the article will be rejected.
Reviewer 2 Report
All Reviewer's Suggestions have been implemented. Therefore, the paper deserves publication.
Author Response
Thank you.